# Research on Theory and a Performance Analysis of an Innovative Rehabilitation Robot

**Junyu Wu** **, Yubin Liu \*, Jie Zhao, Xizhe Zang and Yingzi Guan**

Mechanical and Electrical Engineering, Harbin Institute of Technology, Harbin 150001, China;
21b908013@stu.hit.edu.cn (J.W.); jzhao@hit.edu.cn (J.Z.); zangxizhe@hit.edu.cn (X.Z.); guanyz@hit.edu.cn (Y.G.)
**\*** Correspondence: liuyubin@hit.edu.cn

**Abstract:** This paper presents an innovative application of a 6-DOF robot in the field of rehabilitation training. This robot operates in a parallel fashion for lower limb movement, which adopts a new structure that can help patients to carry out a variety of rehabilitation exercises. Traditional parallel robots, such as the Stewart robot, have the characteristics of strong bearing capacity. However, it is difficult to achieve high-speed, high-acceleration and long journey movement. This paper presents a new robot configuration that can address these problems. This paper also conducts an all-around characteristic analysis of this new parallel robot, including kinematics, dynamics and structure, to better study the robot and improve its performance. This paper optimizes an algorithm to make it more suitable for rehabilitation training. Finally, the performance improvements brought by optimization are verified by simulations.

**Keywords:** parallel robot; washout algorithm; performance analysis; rehabilitation training

## 1. Introduction

Due to an aging population and increases in the number of stroke patients, physical disabilities and other problems, the number of physically unstable people is growing. Human balance disorders are not only a focus of current research in the medical field but also an urgent problem to be solved. A large portion of instability problems can be attributed to abnormalities in the human motion sensing system, especially the degradation or loss of the vestibular sensing system [1]. The main function of a robot is to help patients train through motion simulation. By compelling patients to carry out various rehabilitation exercises, they are constantly stimulated to produce vestibular sensation to guide the brain to actively integrate and improve the ability of various motion sensing organs [2]. Another function occurs under the auxiliary protection of a suspension device; the robot can be used to create an unstable simulation environment. We can then obtain a patient's fall information for a subsequent evaluation of the type or degree of balance disorder.

This paper introduces the structure and advantages of this new robot. The characteristics of the new robot are analyzed. To realize a variety of motions in a limited range of motion, this paper uses a washout control algorithm. To overcome the limitations of the classic washout algorithm, the MOGA (Multi-Objective Genetic Algorithm) algorithm is used to optimize the filter parameters, find the most suitable parameters, and improve the fidelity of motion simulation.

## 2. Mechanisms of Lower Limb Rehabilitation Robots

### 2.1. Research Status

At present, the research on rehabilitation training robots is mainly divided into two categories: wearable exoskeleton robots and end robots. Lokomat [3] and Bleex [4] are notable exoskeleton robots. The Lokomat lower limb rehabilitation robot, developed by the Hocoma Company in Zurich, Switzerland, is based on a traditional weight reduction

support system which is installed outside the patient's lower limb to provide driving force and coordinate with a treadmill to assist in movement, as shown in Figure 1. Recently, the University of Twente in the Netherlands has developed a new robot, Lopes, which is used for gait training of patients with balance disorders. It is composed of three rotating joints and a translation component, in which the rotating joints are driven and the translation component realizes the function of the human hip joint. The German Lokohelp team has developed a robot for lower limb rehabilitation training. Its structure is similar to that of Lokomat. It is also based on weight loss supporting the exoskeleton and treadmill, but it transmits the movement of the treadmill to the treadmill, which drives the movement of human lower limbs for gait training [5]. Although exoskeleton robot equipment can provide better training for patients, these robots are bulky, expensive, complex to operate and usually need more than two caregivers for implementation, so the cost of lower limb rehabilitation training is substantial.

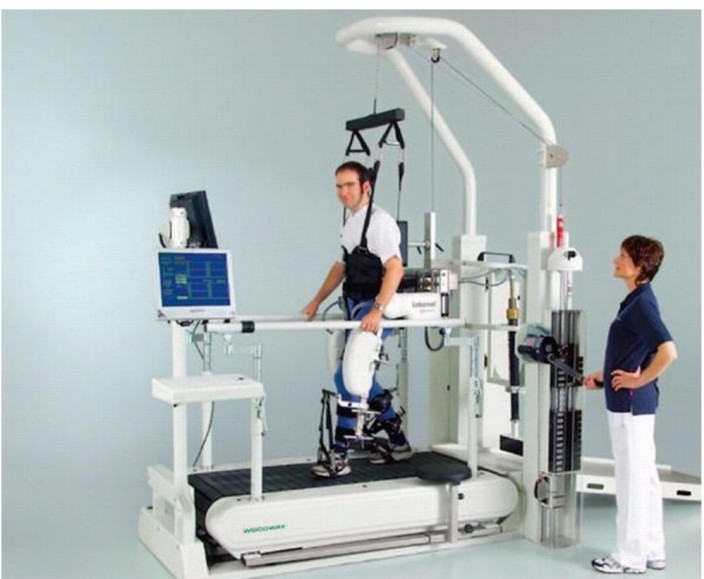

**Figure 1.** Typical exoskeleton robots: Lokomat.

There are two types of end robots: pedal robots and platform robots. A treadmill-based robot usually consists of a body weight reduction support system to assist patients in walking training on a treadmill. A pedal rehabilitation robot fixes the patient's foot on the robot pedal and is controlled to drive the lower limbs to simulate different gait movements. In recent years, rope-driven parallel robots have been used in rehabilitation training [6]. Suzhou University in China has designed an 8-rope-driven 6-DOF parallel rehabilitation robot for gait rehabilitation training. As shown in Figure 2, the robot can effectively provide weight reduction in the process of rehabilitation training and can protect the patient's hip position within a controllable range while realizing multiple degrees of freedom in hip movement [7].

Yeongmin Kim of the Department of mechanical engineering at Xijiang university designed a rehabilitation training robot, EXOWheel [8], for strength training for specific muscle groups of lower limbs, as shown in Figure 3. Through the optimization method, the seven muscles are taken as a function of the position in the predefined circular trajectory, and the patient moves his lower limbs along the predetermined circular trajectory, which can maximize the efficiency of specific muscle groups.

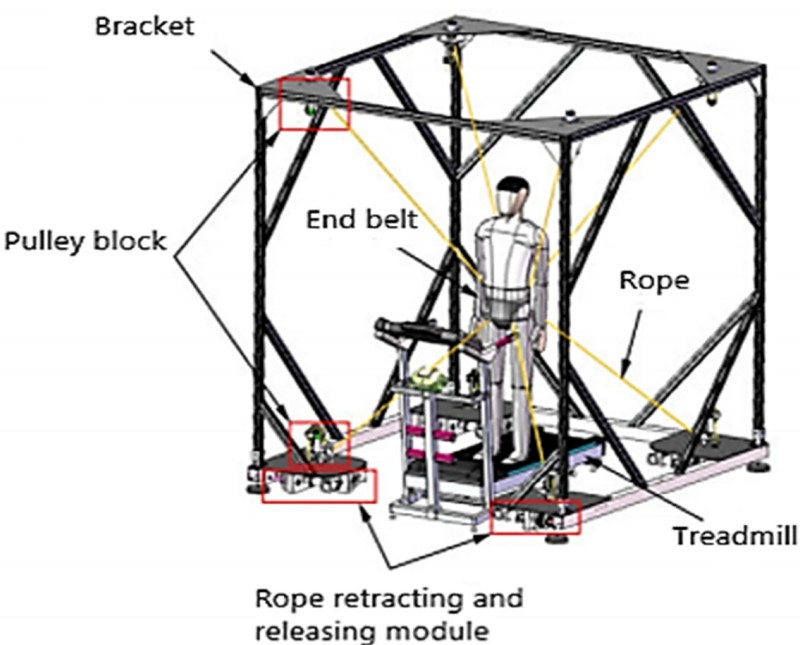

**Figure 2.** Rope-driven parallel robot.

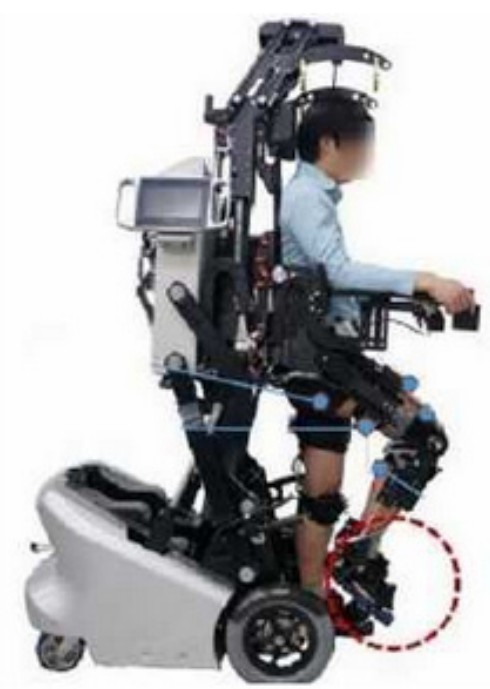

**Figure 3.** EXOWheel.

Trinnachoke Eiammanussakul and Viboon Sangveraphunsiri of the mechanical engineering department at Chulalongkorn University designed a sitting rehabilitation training robot for patients with severe lower limb disorders who cannot walk [9], as shown in Figure 4. It is mainly composed of mechanical leg, balance device and control parts. The range of motion is determined by the brake installed at the joint.

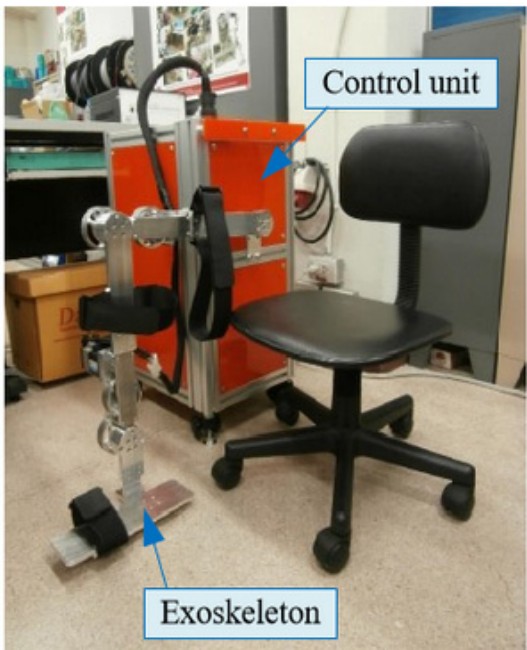
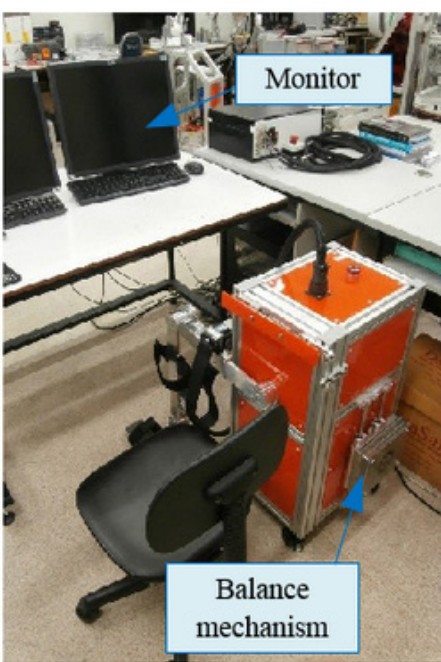

**Figure 4.** A sitting rehabilitation training robot.

## 2.2. The Proposed Structural Form

A new 6-DOF parallel platform with a large load and high acceleration is proposed in this paper. The structural diagram of the robot is shown in Figure 5 below. The platform is composed of two identical 6-SSP parallel robots, which are symmetrically mirrored. One of the devices is mainly composed of three parts: a foundation platform, a transmission chain and a moving platform. The foundation platform is fixed, which mainly provides for support and the placement of moving transmission parts. The transmission form consists of six linear moving pairs, 12 spherical joints, and six connecting rods. Each linear pair can be completed in the form of a guide rail and slider. Six linear pairs are arranged in a vertical plane in pairs, which are divided into three groups. Each group is at the same height, and the three guide rails are equally spaced and parallel to each other. Six linear pairs are used as active pairs to complete one-dimensional sliding in the horizontal direction. The motion and power are transmitted to the moving platform. The moving platform has six degrees of freedom. The human foot steps on the moving platform. Each moving platform acts as the support point of one foot. The patient stands on the platform with the help of auxiliary devices, such as a suspension device. The two moving platforms cooperate with each other to actively drive human movement and complete a variety of patient motion simulations.

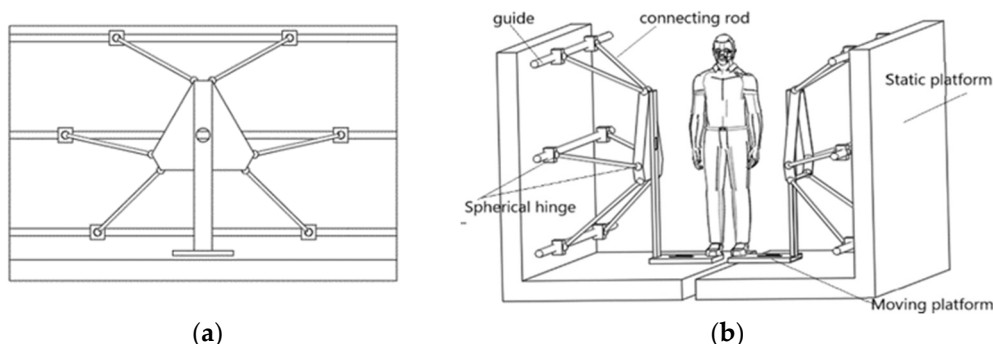

(**a**)          (**b**)

**Figure 5.** Schematic diagram of robot structure. (**a**) Structure diagram of unilateral robot. (**b**) Structure diagram of the whole machine.

### 3. Characteristic Analysis

*3.1. Kinematics Analysis*

A kinematics analysis describes the motion relative to a fixed-reference Cartesian coordinate system in which the forces and moments leading to the motion of the structure are not considered. This paper shows the establishment of forward and inverse kinematics models. Some of the results are displayed visually in Figure 6.

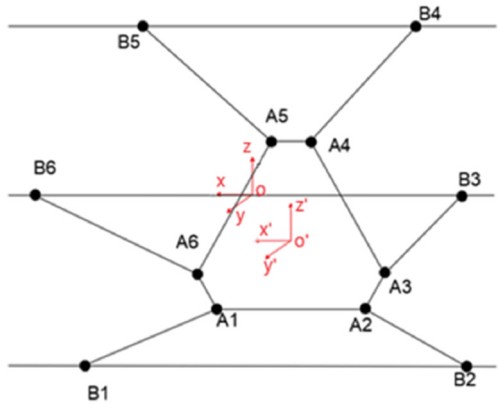

**Figure 6.** Schematic diagram of the mechanism.

### 3.1.1. Forward Kinematics

The forward kinematics problem for a robot resolves the homogeneous transformation matrix (three translation transformations and three rotation transformations) with known rod parameters and joint motion parameters of the robot.

Assume that the initial pose of the moving platform is $(X_{P0}, Y_{P0}, Z_{P0}, \gamma_0, \beta_0, \alpha_0)$. The forward kinematics model can be easily established by using the formula of the distance between two points and rod length. Assuming that the rod length is $L_i$ and the homogeneous transformation matrix is T, the calculation formulas are as follows:

$$T = \begin{bmatrix} n_x & s_x & a_x & x_p \\ n_y & s_y & a_y & y_p \\ n_z & s_z & a_z & z_p \\ 0 & 0 & 0 & 1 \end{bmatrix}$$

$$L_i = \sqrt{(X_{Ai} - X_{Bi})^2 + (Y_{Ai} - Y_{Bi})^2 + (Z_{Ai} - Z_{Bi})^2} \tag{1}$$

$$n_x^2 + n_y^2 + n_z^2 = 1 \tag{2}$$

$$s_x^2 + s_y^2 + s_z^2 = 1 \tag{3}$$

$$a_x^2 + a_y^2 + a_z^2 = 1 \tag{4}$$

$$n_x \cdot a_x + n_y \cdot a_y + n_z \cdot a_z = 0 \tag{5}$$

$$s_x \cdot a_x + s_y \cdot a_y + s_z \cdot a_z = 0 \tag{6}$$

$$n_x \cdot s_x + n_y \cdot s_y + n_z \cdot s_z = 0 \tag{7}$$

The homogeneous transformation matrix T can be obtained by the simultaneous establishment of 12 formulas. Due to there being only six independent variables, the forward kinematics problem solves the six element nonlinear equations.

For nonlinear equations and systems of equations, since the analytical solution cannot be obtained, a numerical solution based on the quasi-Newton iterative method is used to obtain the approximate solution with the required accuracy.

### 3.1.2. Inverse Kinematics

The inverse problem of the robot resolves the joint motion parameters (the movement of six moving pairs) through the known robot member parameters and homogeneous transformation matrix T.

The coordinate systems o'-x'y'z' and o-xyz are established on the upper and lower platforms, respectively. The upper platform is the moving platform, and the lower platform is the foundation platform. The coordinates of points in space in the static coordinate system are $A_i(X_{Ai}, Y_{Ai}, Z_{Ai})(i = 1, 2 \ldots 6)$. When the pose data of the upper platform ($X_p$, $Y_p$, $Z_p$, $\gamma$, $\beta$, $\alpha$) are known, the homogeneous transformation matrix T between the two coordinate systems is as shown in Formula (8), and the rotation transformation matrix $R$ is as shown in Formula (9).

$$T = \begin{bmatrix} & & & Xp \\ & R & & Yp \\ & & & Zp \\ 0 & 0 & 0 & 1 \end{bmatrix} \tag{8}$$

$$R = \begin{bmatrix} n_x & s_x & a_x \\ n_y & s_y & a_y \\ n_z & s_z & a_z \end{bmatrix} = \begin{bmatrix} c\gamma c\beta & -s\gamma c\beta & s\beta \\ s\gamma c\alpha + c\gamma s\alpha s\beta & c\gamma c\alpha - s\gamma s\beta s\alpha & -c\beta s\beta \\ s\gamma c\alpha - c\gamma s\beta c\alpha & c\gamma s\alpha + s\gamma s\beta c\alpha & c\beta c\alpha \end{bmatrix} \tag{9}$$

If the coordinate of $Ai$ in the moving coordinate system is $A_i'$ ($X_{Ai}', Y_{Ai}', Z_{Ai}'$), then the homogeneous transformation is $A_i = T \cdot A_i'$. If the $A_i$ coordinates of points in space and the pose data of the upper platform are known, the coordinate of point $Bi$ can be obtained from the space vector relationship.

In summation:

$$L_i = \sqrt{(X_{Bi} - X_{Ai})^2 + (Y_{Bi} - Y_{Ai})^2 + (Z_{Bi} - Z_{Ai})^2} \tag{10}$$

When the initial position coordinate of the moving platform relative to the static coordinate system is (0, 0, 600 mm), the attitude is roll angle = 20°, and the yaw angle and pitch angle are 0°. The positions of the six sliders in the X-direction are [885.05, 1166.37, 831.88, −831.88, −1166.37, −885.05] (mm). The pose of the moving platform is shown in Figure 7.

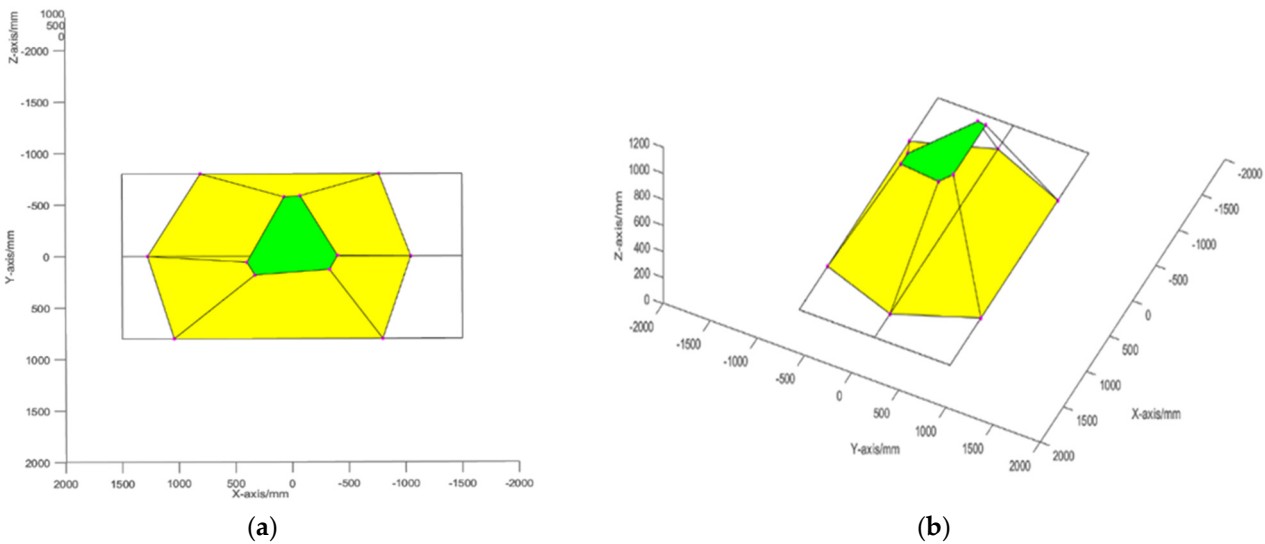

(**a**)          (**b**)

**Figure 7.** Pose of the robot when the roll angle = 20 degrees. (**a**,**b**) show the results from different perspectives.

When the initial position coordinate of the moving platform relative to the static coordinate system is (0, 0, 600 mm), the yaw angle = 20°, and the roll angle and pitch

angle are 0°. The positions of the six sliders in the X-direction are [913.31, 1088.87, 917.69, −896.16, −1253.66, −634.91] (mm). The pose of the moving platform is shown in Figure 8.

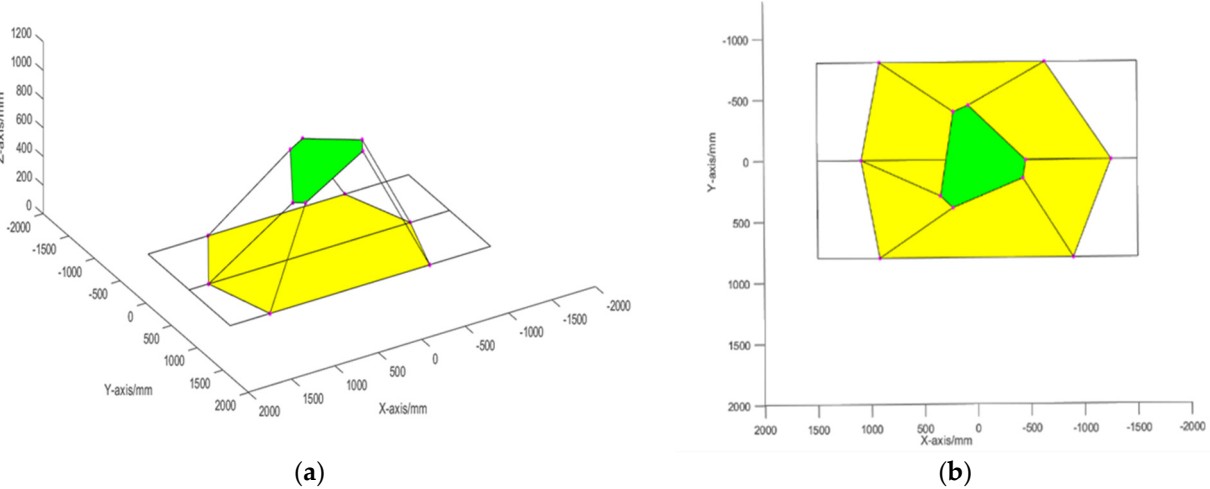

(**a**)　　　　　　　　　　　　　　　　　　　　　　　　　(**b**)

**Figure 8.** Pose of the robot when the yaw angle = 20 degrees. (**a**,**b**) show the results from different perspectives.

### 3.2. Dynamics

The purpose of studying dynamics is to further improve the model and study the relationship between the driving force of each joint and the external force during the movement of the robot. In the study of robot kinematics there are two aspects, positive solution and inverse solution, related to the dynamics. For a complete system, the Lagrange dynamic equation expressed in generalized coordinates usually refers to the second kind of Lagrange equation, which is first derived from the French mathematician Lagrange, and the following Formula (11) is typically used [10].

$$\mathrm{F_i} = \frac{d}{dt}\frac{\partial L}{\partial \dot{q}_i}\frac{\partial L}{\partial q_i} \tag{11}$$

where T is the kinetic energy expressed by each generalized coordinate $q_i$ and each generalized velocity $\dot{q}_i$ of the system, and Fi is the generalized force corresponding to $q_j$·N (3n − k) which represents the degrees of freedom of the complete system. N is the prime number of the system and k is the number of holonomic constraint equations.

Due to space limitations, the derivation and establishment of the Lagrange dynamic equation of the robot are not shown. Some results and analyses of dynamic simulations are shown below. The results show two dynamic simulations.

### 3.2.1. First Dynamic Simulation

Parameter setting: The main structural parameters of the platform include a guide rail spacing of 800 mm and a rod length of 1000 mm. Let the platform move according to the specified acceleration command. The acceleration is a step signal, which changes from 5 m/s² to 0.5 m/s² in 0.1 s. The direction is the positive direction of the Z-axis. Figure 9 shows this acceleration signal. Figure 10 shows the dynamic simulation results.

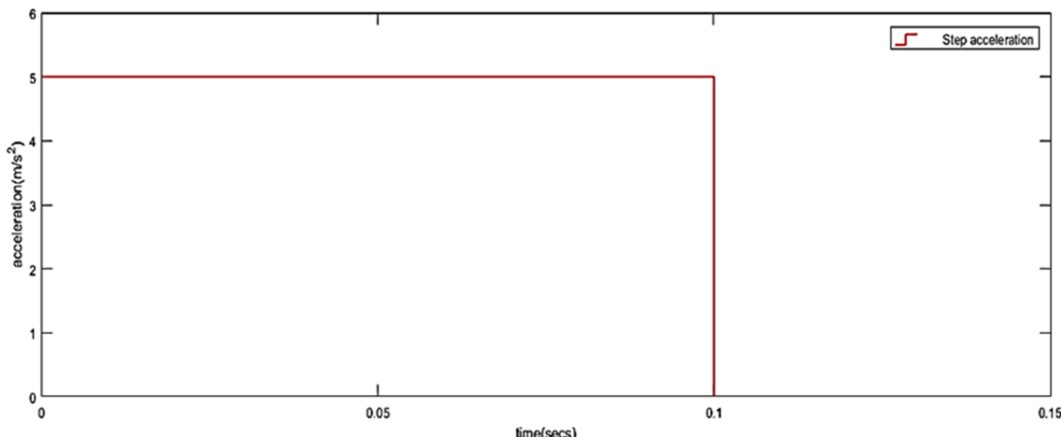

**Figure 9.** Step acceleration signal.

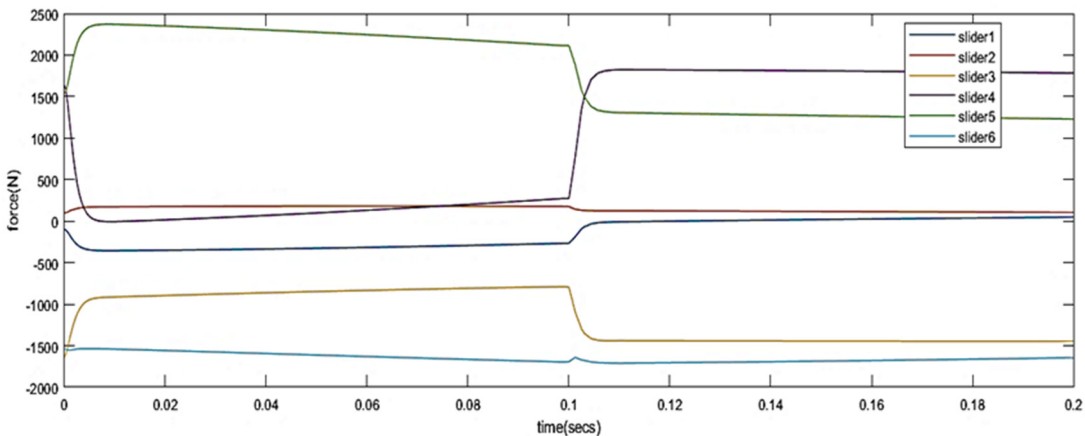

**Figure 10.** First dynamic simulation result.

### 3.2.2. Second Dynamic Simulation

The structural parameters are the same as before. The acceleration is also a step signal, which changes from 5 m/s$^2$ to 0.5 m/s$^2$ in 0.1 s. However, the direction is the positive direction of the Y-axis. Figure 11 shows the dynamic simulation results.

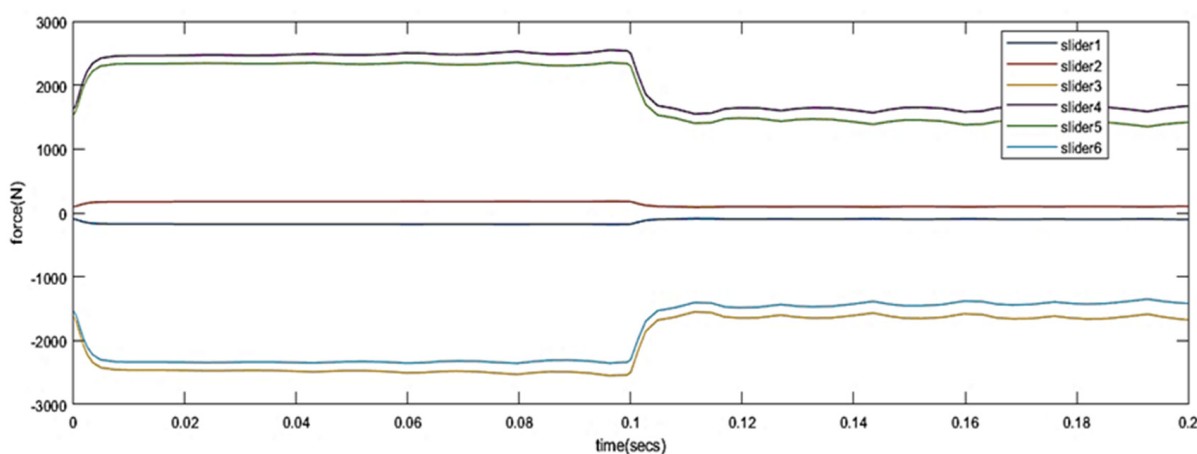

**Figure 11.** Second dynamic simulation result.

The simulation results are affected by the length of the connecting rod, the distance between the guide rails, the initial position of the end platform and other factors. The

simulation results show that the driving force on each linear joint is between 500 N and 3000 N without considering friction.

### 3.3. Finite Element Analysis

Many parallel robot performances are closely related to their own structural characteristics, such as natural frequency, vibration mode and damping. For example, if the natural vibration frequency of the robot is close to the external interference vibration source, resonance will occur, which may cause large deformation and stress within the robot's own structure [11]. When considering the application of rehabilitation training for patients with balance disorders, these consequences are unimaginable. Therefore, it is very important to carry out modal analysis and avoid resonance. There are three methods used to study the structural dynamic characteristics and obtain the corresponding structural dynamic parameters of the robot: experimental methods, theoretical analysis methods and software simulation analyses. For simple structures, we can obtain the corresponding structural dynamic parameters through manual calculation, but for complex structures many parameters are coupled with each other, which is difficult to model and calculate, so this method has certain limitations. With the help of structural dynamic analysis software and the powerful computing function of computers, we can establish the relationship between complex structures and mechanics, and calculate and optimize structural dynamic parameters.

#### 3.3.1. Structural Static Analysis

Structural static analysis mainly studies the strength, stiffness and bearing capacity of the structure when the parallel robot is subjected to static load and finds the weak links in the structure. The design of dangerous parts should be mainly considered. At the same time, in the early stage of ensuring reliability, we try to reduce the weight of the robot and realize light weight and high speed. A load of 250 kg was applied to the center of the moving platform. The following Figure 12 shows the relevant results.

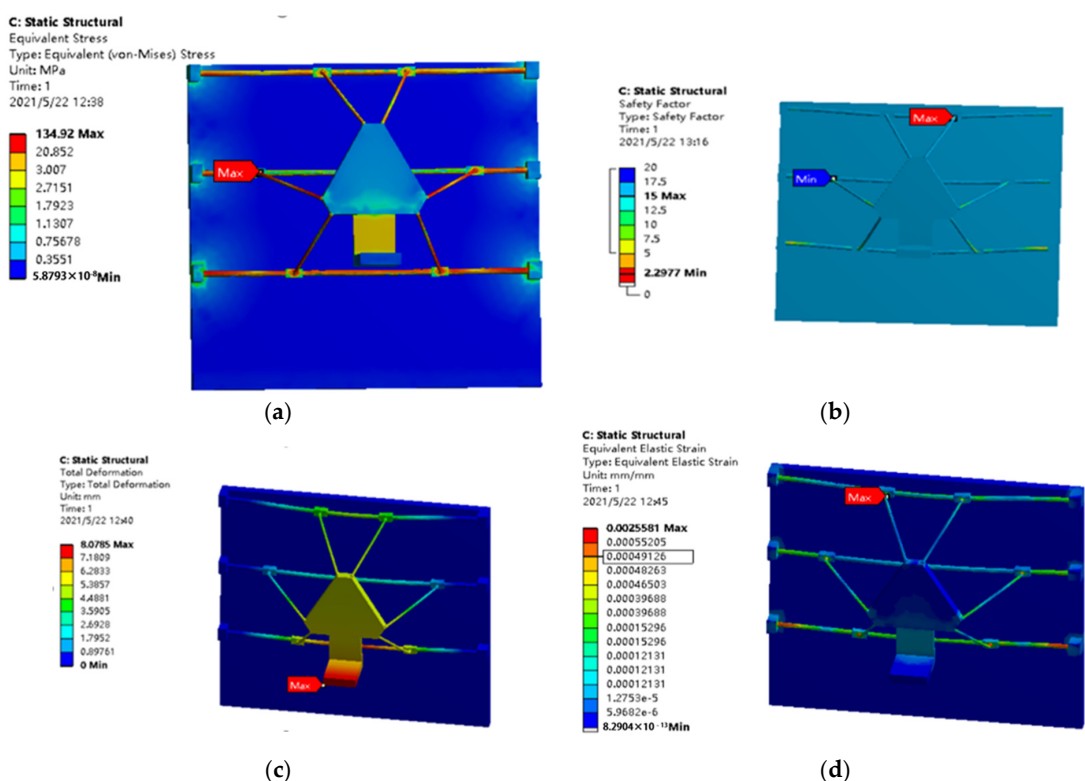

**Figure 12.** Structural static simulation results: (**a**) equivalent force nephogram; (**b**) safety factor nephogram; (**c**) total deformation nephogram; (**d**) strain nephogram.

By analyzing the simulation results it can be seen that the maximum stress value is 134.92 MPa, and the maximum stress appears at the position where the rod is connected with the sliding block. Therefore, we should focus on the strength of the hook hinge device. Compared with other parts, the connecting rod is a dangerous part. The maximum stress on the rod is 75.319 Mpa, the average stress is 17.972 Mpa and the minimum safety factor is 3.33.

### 3.3.2. Modal Analysis

Modal analysis is mainly used to understand the structural dynamic characteristics of the parallel robot, obtain the structural dynamic parameters, avoid external interference sources that may cause structural resonance, and help us understand the stiffness performance of the robot and optimize its structure [12].

There is no load in modal analysis because the dynamic characteristics of the structure are only related to its own structural materials. We carry out modal analysis on the whole robot and select the most typical pose of the robot, that is, the initial position (the initial position here refers to the initial position of the robot in the washout algorithm). All structural systems have distributed mass, so they are infinite degrees of freedom systems, thus the order of modes is infinite. Since the robot has six degrees of freedom before training, it has six degrees of freedom for rehabilitation analysis. The vibration mode diagram of the first six modes is shown in Figure 13.

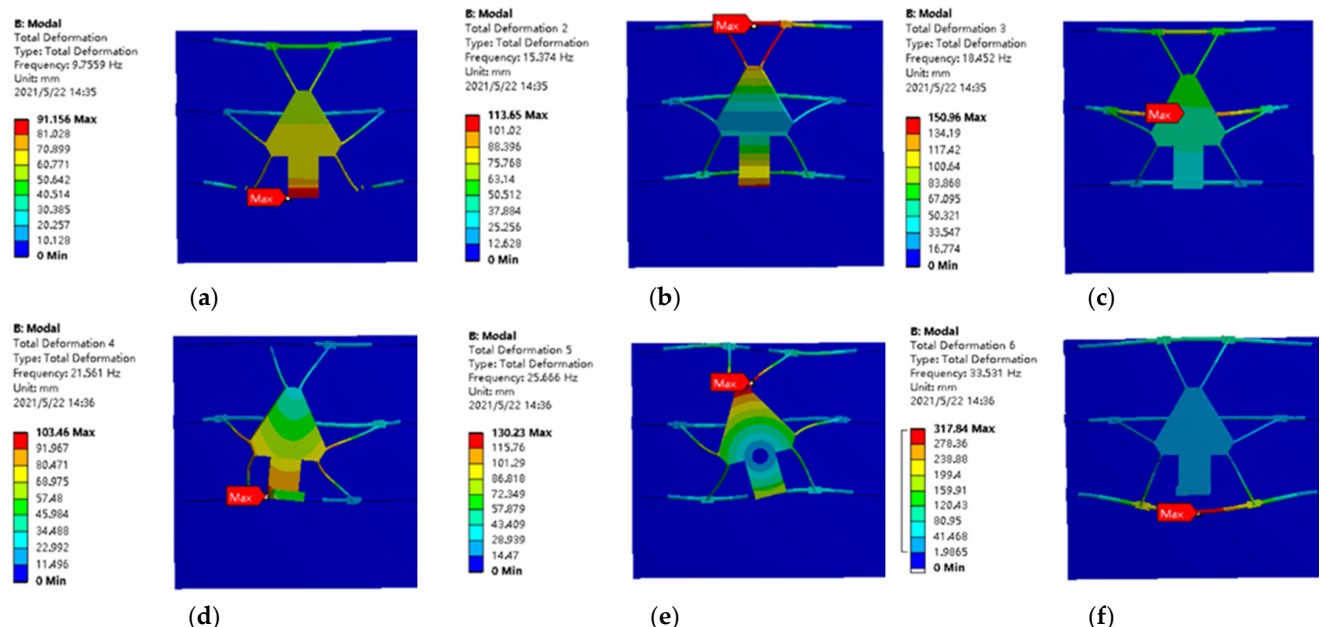

**Figure 13.** Modal simulation results: (**a**) first-order mode; (**b**) second-order mode; (**c**) third-order mode; (**d**) fourth-order mode; (**e**) fifth-order mode; (**f**) sixth-order mode.

The results show that the first three modes are mainly manifested in the pitching motion of the moving platform, the fourth mode is manifested in the yaw motion of the platform, the fifth mode is realized as the rolling motion of the platform, and the sixth mode is mainly the up and down vibration and torsion of the guide rail. In all modes, the guide rail exhibits large vibration and torsion, indicating that the stiffness of the guide rail is poor. Thus, to simplify the model, the guide rail is replaced by a column with a diameter of 30 mm, and the installation and fixation method is fixed at both ends, which is inconsistent with the actual installation form of the guide rail. In actual work, the external interference source will not cause such a violent resonance phenomenon. The connecting rod also has a small amount of distortion, which shows that although the rod meets the strength requirements, its stiffness should also be appropriately enhanced. Therefore, the

cross-sectional area of the rod should be appropriately increased, and a material with greater stiffness should be selected. Table 1 shows the natural frequencies of each mode.

**Table 1.** First 6 natural frequencies.

| Order | Frequency (Hz) |
|:-----:|:--------------:|
| 1 | 9.7559 |
| 2 | 15.374 |
| 3 | 18.452 |
| 4 | 21.561 |
| 5 | 25.666 |
| 6 | 33.531 |

## 4. Research on the Control Algorithm

### 4.1. Introduction to the Washout Algorithm

The washout algorithm is a key algorithm for various motion simulators and is used to simulate human vestibular somatosensory perception. The vestibule is an organ located in the inner ear of the human body. It senses the changes in acceleration and angular velocity of the head and transmits the stimulation brought by movement to the central nerve to cause corresponding sensation. However, the human motion perception system has defects. When the linear acceleration and rotational angular velocity are less than the human perception threshold, people cannot feel the corresponding stimulation. In motion simulation, the primary solution is the limited motion space of the platform. According to the defects of the human perception system, the washout algorithm washes out the platform to the initial position when the human body is not aware enough to realize the vestibular perception applied by the robot in the limited workspace and infinite simulation motion environment [13].

The washout algorithm is mainly composed of four parts: a tilt coordination channel, a coordinate transformation link, an acceleration high-pass channel and an angular velocity low-pass channel. In addition to the coordinate transformation link, the other three links are composed of a series of filters, and the function of each filter is to wash out the motion instructions (the motion expected to be perceived by the human body) into the motion of the platform, so the parameters of each filter are very important for the fidelity of motion simulation. Filter parameters include order, natural frequency and damping ratio. In this paper, the high-pass filter is selected as the third-order filter, and the tilt coordination link and angular velocity low-pass link are selected as the second-order filter. The transfer function of the high-pass filter [14] is:

$$G_1(s) = \frac{s^2}{s^2 + 2\xi\omega_a s + \omega_a{}^2} \cdot \frac{s}{s + \omega_a'} \tag{12}$$

In the tilt coordination channel, the transfer function of the second-order filter is:

$$G_2(s) = \frac{s^2}{s^2 + 2\xi\omega_h s + \omega_h{}^2} \tag{13}$$

The classical washout algorithm is flawed. Because the classical washout algorithm was first applied to the simulator of automobile or aircraft. It simulates continuous acceleration in one direction similar to aircraft take-off or car turning. When the rehabilitation training robot assists the patient in walking movement, the acceleration of the patient's movement is more complex, and the movement can be ideally regarded as the periodic change of acceleration. Therefore, the classical washout algorithm has limitations in this application scenario. The key of classical washout algorithm is the filter parameters in each channel. Therefore, we use an optimization method to find the filter parameters that are most suitable for the application scenario of rehabilitation training. The optimization will be described in detail below.

### 4.2. Parameter Optimization of Washout Algorithm

In this paper, a multiobjective genetic algorithm is used to find the optimal filter parameters and enable the most realistic motion feeling. A multiobjective genetic algorithm is an improved evolutionary algorithm based on a genetic algorithm, which resolves the multiobjective function problem with better solution results [15]. There are three optimized target variables [$\omega_a$, $\omega_a'$, $\omega_h$]. The purpose of optimization is to bring a realistic sense of motion to the rehabilitation training robot. To quantitatively describe the fidelity of motion simulation, a semicircular canal model and otolith model of the human vestibule are introduced. When the human body directly senses the acceleration information, the acceleration signal is directly input to the otolith model. The real somatosensory acceleration information can be obtained, which is called the ideal perceived acceleration. If the acceleration information is input to the washout algorithm as a command, the acceleration information generated by the human after the movement of the moving platform is called the actual perceived acceleration. The difference between the two acceleration values can be obtained, that is, the human perceived acceleration error. Similarly, the angular velocity error of the human body can also be obtained. In this way, an objective function of optimization is derived.

The transfer function of the otolith is shown in Formula (14):

$$G_{oto}(s) = \frac{\hat{f}}{f} = \frac{k(\tau_a s + 1)}{(\tau_l s + 1)(\tau_s s + 1)} \tag{14}$$

($\tau_a$, $\tau_l$, $\tau_s$: otolith time constant, long time constant, short time constant, dimensionless)
The transfer function of the semicircular canal is shown in Formula (15):

$$G_{sc}(s) = \frac{\hat{\omega}}{\omega} = \frac{T_a T_l s^2}{(T_l s + 1)(T_s s + 1)(T_a s + 1)} \tag{15}$$

($T_a$, $T_l$, $T_s$: semicircular canal time constant, long time constant, short time constant, dimensionless)

The X-direction (axial) coefficient of the otolith model is shown in Table 2 below:

**Table 2.** Coefficient of otolith model.

| Parameter | k | $\tau_a$ | $\tau_l$ | $\tau_s$ |
|---|---|---|---|---|
| X-direction | 0.4 | 13.2 | 5.33 | 0.66 |

The X-direction (rolling) coefficient of the semicircular canal model is shown in Table 3 below.

**Table 3.** Coefficient of the semicircular canal model.

| Parameter | $T_a$ | $T_l$ | $T_s$ |
|---|---|---|---|
| X-direction | 6.1 | 0.1 | 30 |

However, a small somatosensory error will inevitably lead to a larger motion of the platform and a longer motion time of the platform. The motion may easily exceed the limit position requirements of the platform and may also lead to phase delay when the platform returns, which will affect the next motion simulation. Therefore, the displacement movement of the platform should be as small as possible, which should also be a goal of optimization. Therefore, there are two optimization objectives. One is to reduce the human body sensing error, including the human body sensing acceleration error and the body sensing angular velocity error. The other is to minimize the motion of the washed out platform, including the displacement of the platform and the angular velocity of the platform.

For the multiobjective optimization problem, the weighting method is used to give the target value of each objective weight according to its importance, multiply the target function value and the weight, and then add linearly to transform it into a single objective function. However, this weight is not a fixed coefficient. In the MOGA, the weight is a function related to the number of individuals dominating other individuals. Moreover, there is a certain gap in the magnitude of the two objective functions. The sensitivity of the two objectives to the design variables is different, which affects the stability and convergence speed of the search process. Its scale should be processed, that is, normalized [16]. The optimization is completed in MATLAB.

In summary, the mathematical model of the optimization objective is given as follows: (16) to (19);

$$L[e_{ax}(t)] \quad [a_x(t)](G_{oto}(s) - G_{oto}(s) \times G_{ahx}(s)) \tag{16}$$

$$L[e_{wx}(t)] \quad [\omega_x(t)](G_{ahs}(s) - G_{ahs}(s) \times G_{wx}(s)) \tag{17}$$

$$L[d_x(t)] \quad G_{oto}(s) \times a_x(t) \times \left(\frac{1}{s^2}\right) \tag{18}$$

$$L[\omega_x(t)] \quad G_{ahs}(s) \times a_x(t) \tag{19}$$

$L[e_{ax}(t)]$—Somatosensory error function of X-direction (longitudinal) acceleration
$L[e_{wx}(t)]$—Somatosensory error function of angular velocity in the X direction (rolling)
$L[a_x(t)]$—Laplace transform of acceleration signal
$L[\omega_x(t)]$—Laplace transform of roll angle velocity
$L[d_x(t)]$—Longitudinal displacement of platform
$G_{ahx}(s)$—X-direction high pass filter
$G_{wx}(s)$—X-direction tilt coordinated filter

The input signal is a periodic acceleration square wave signal with a period of 2 s, amplitude of 2 m/s$^2$ and duration of 10 s. Figure 14 shows this signal.

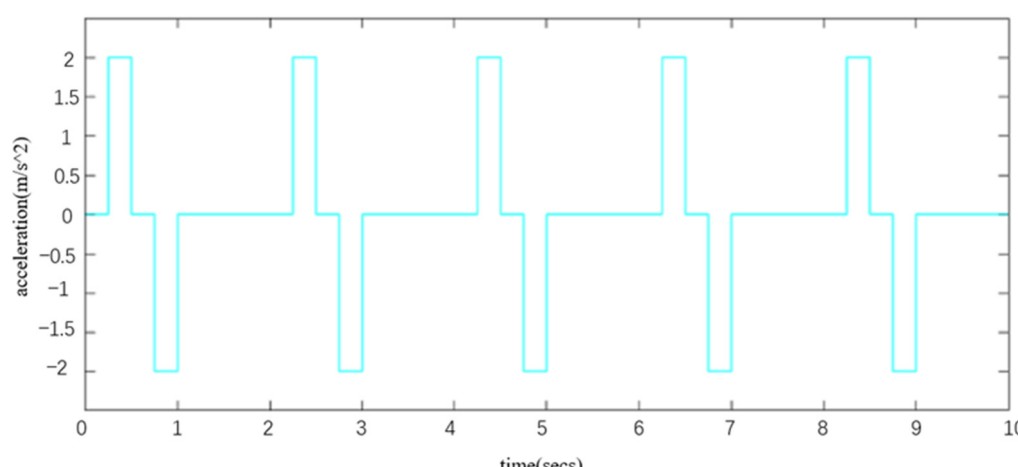

**Figure 14.** Input acceleration signal.

We analyze the simulation results of human sensory error before and after optimization, use the washout algorithm after parameter optimization to reduce the error of human somatosensory acceleration and angular velocity, and make the somatosensory simulation more realistic. The displacement of the platform washed out before and after optimization is significantly improved. Not only is the amplitude of displacement reduced, but the action range of the platform decreases, the platform is more likely to work within the safe formation range, and the platform returns to its original position faster which can prepare for the next simulation faster. The angular velocity of the platform before and after optimization does not exceed the required threshold, but the situation after optimization is slightly improved, the amplitude of angular velocity decreases, and the action ampli-

tude quickens. In conclusion, the optimized parameters comprehensively improve the performance of platform motion simulation. Figure 15 show the motion information and somatosensory error of the platform before and after optimization.

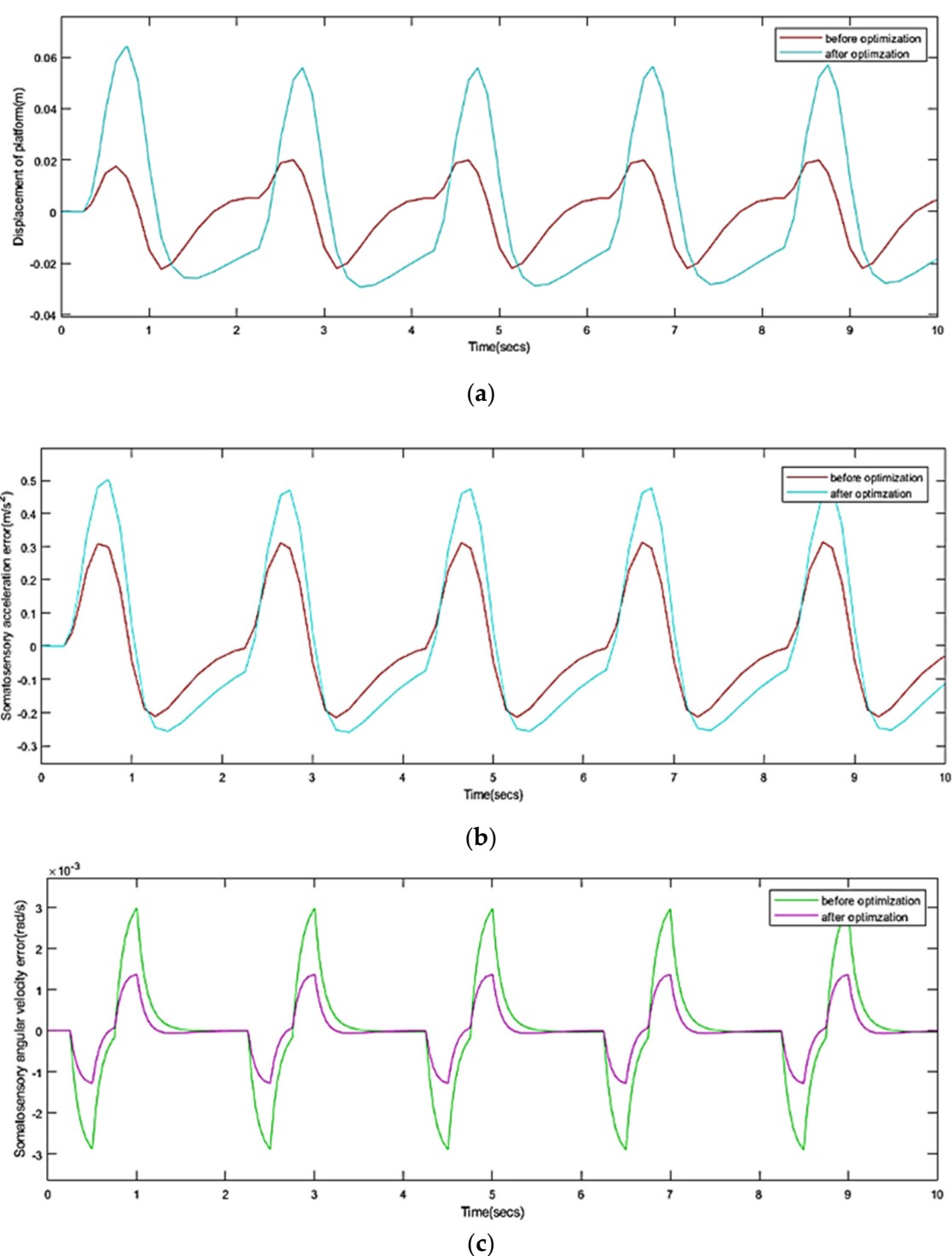

**Figure 15.** *Cont.*

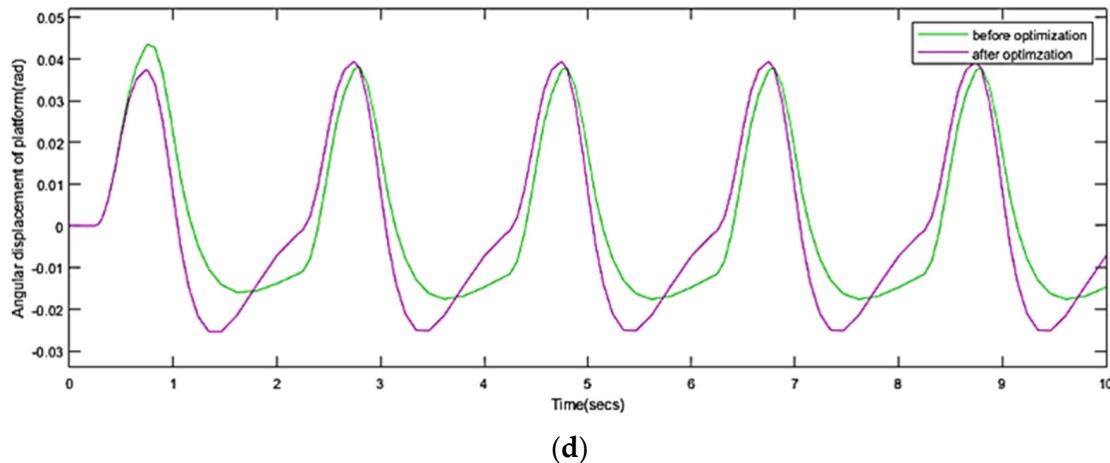

**(d)**

**Figure 15.** Value of optimization target before and after optimization: (**a**) displacement; (**b**) somatosensory acceleration error; (**c**) angular displacement; (**d**) somatosensory angular velocity error.

### 4.3. Simulation Verification of Optimization Results

First, a three-dimensional simulation model of the robot is created through the Simscape Library in Simulink. The framework of the washout algorithm is then built through the Simulink and m functions. The relevant parts of the washout algorithm and platform simulation model are connected in series, and the platform motion information washed out by the washout algorithm is used as the input of the platform simulation model to control the motion of the platform. The input acceleration is a step acceleration signal in the Y-direction, and the size is 0.2 $g$ ($g$ is the gravitational acceleration). At 10 seconds, the acceleration suddenly changes to 0. In this way, we obtain not only the motion information of the platform but also the displacement, velocity and joint driving force of the linear pair. Figure 16 shows the optimized results.

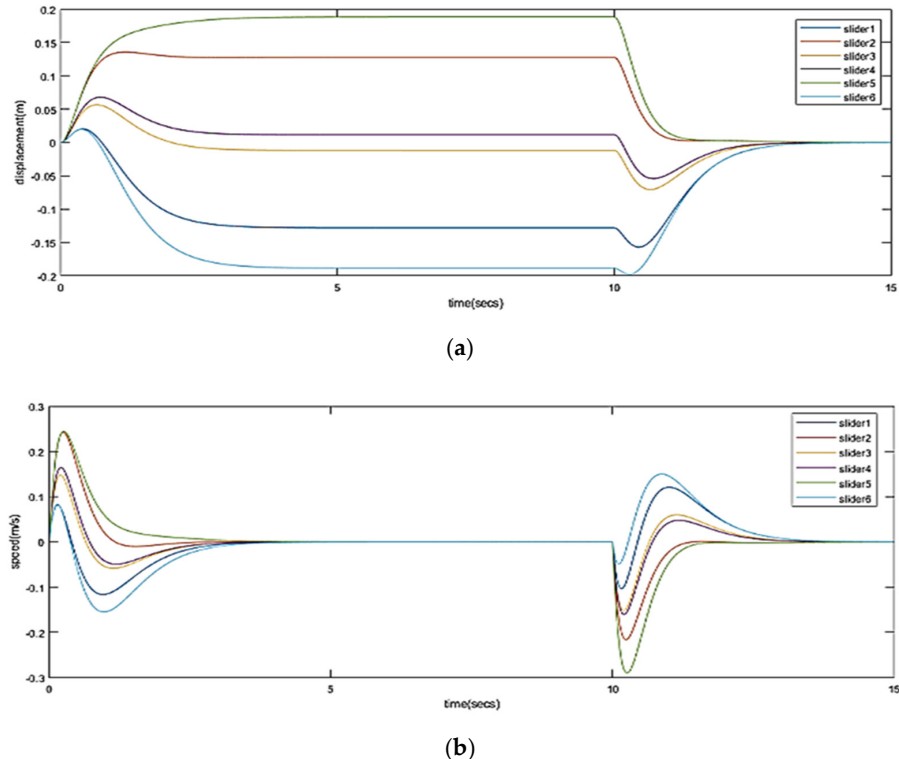

**(a)**

**(b)**

**Figure 16.** *Cont*.

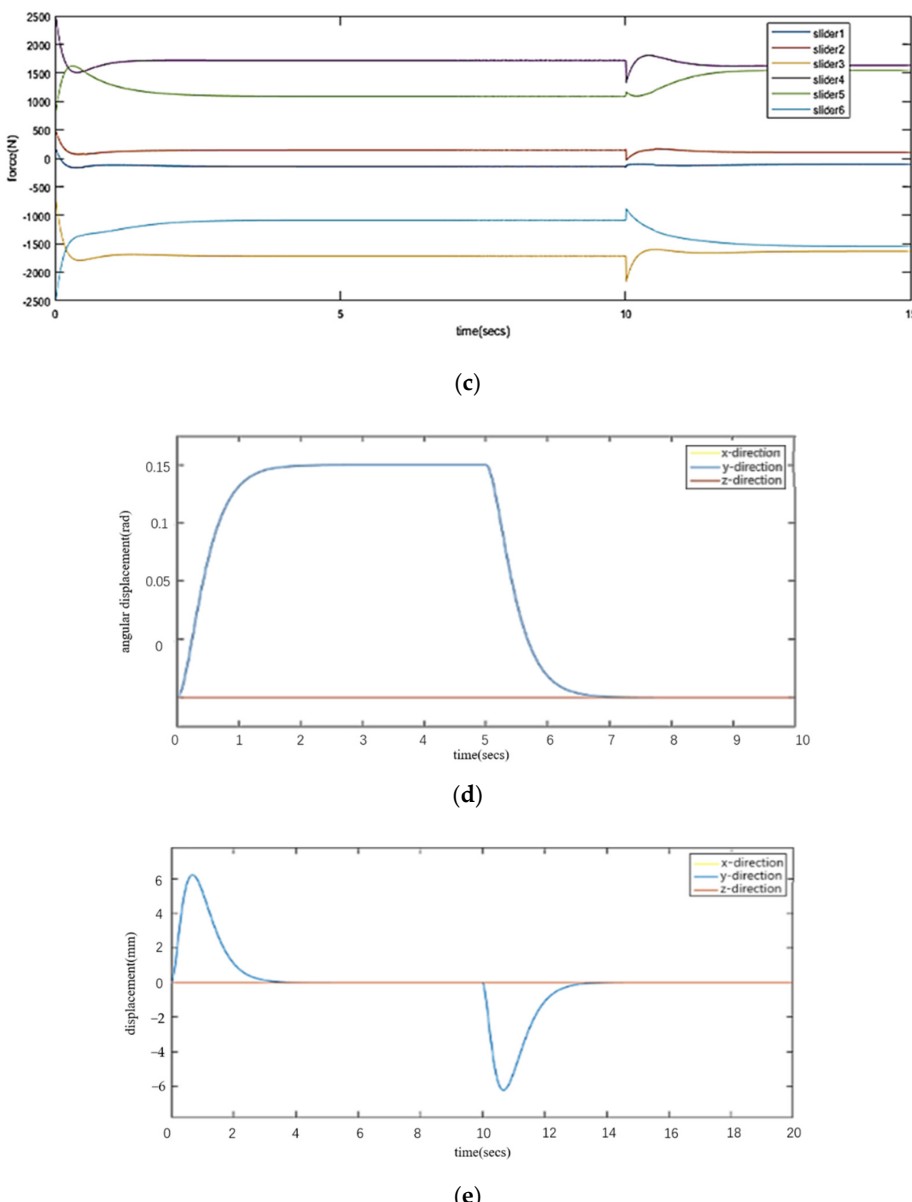

(c)

(d)

(e)

**Figure 16.** The optimized results are used to simulate the motion of the platform so as to obtain the motion and force of the platform and linear pair: (**a**) displacement of linear pair; (**b**) speed of linear pair; (**c**) drive of linear pair; (**d**) angular displacement of the platform; (**e**) displacement of platform.

## 5. Discussion

Result analysis: The actual motion of the platform and the measured motion parameters and force parameters of the platform and each linear pair are in line with expectations. When the acceleration of the Y-direction is 0.2 g, the platform will generate an acceleration that gives people the feeling of acceleration at the beginning. However, the range of motion of the platform is limited, and acceleration will not be provided continuously. To ensure that the motion of the platform does not exceed the range, the acceleration of the platform decreases rapidly. To make people feel continuous acceleration, it is necessary to utilize the tilt coordination channel in the algorithm and use the defects in human perception. The human body can perceive acceleration, but it is impossible to know whether acceleration is generated by the component of actual acceleration or gravitational acceleration in this direction. Therefore, at this time, the platform rotates an angle around the Y-axis, making the user's gravitational acceleration produce a component in the Y-direction and then making people feel the continuous acceleration. When the speed command disappears,

the platform will quickly return to its original position to prepare for the next movement. Therefore, the simulation motion of the platform conforms to the analysis.

Within two seconds after the simulation, the platform turns the corresponding angle. The magnitude of this angle can be calculated by the tilt coordination principle, $a = g \times sin(\Phi)$, where $a$ is the acceleration command, $g$ is the gravitational acceleration, and $\Phi$ is the angle that the platform should turn. In this simulation experiment, the calculation formula is shown in Formula (20):

$$\Phi = arcsin\left(\frac{a}{g}\right) \; arcsin(0.2) \; = 0.2 \; rad \tag{20}$$

Therefore, the platform will not rotate after turning 0.2 rad, which can make the human body perceive a continuous and stable acceleration signal. When it reaches ten seconds, the acceleration command becomes 0, and the platform quickly returns to its original position. According to the observed angular velocity signal of the platform, the maximum angular velocity of the platform during the whole process is 0.147 rad/s, which does not exceed the perceived lower limit of diagonal velocity of the human body. Therefore, the rotation of the platform will not be perceived by the human body.

## 6. Conclusions

This paper designs and constructs a robot system to provide somatosensory simulation for patients in the process of rehabilitation training in the medical field. A new 6-SSP parallel robot is proposed. Its main function is to help patients with balance disorders with somatosensory simulation and rehabilitation training. The following is a comparison of the robot in this paper with other lower limb rehabilitation training robots mentioned above in terms of structure. Compared with exoskeleton robots, such as Lokomat and Bleex, this robot has simple structure, low cost, and is safer and more stable. Compared to the end robot, for example, this robot has a larger motion stroke and realizes high-speeds and high acceleration more easily. Compared to the EXOWheel, the robot can exercise all the moving muscle groups of the lower limbs. The robot in this paper has a long stroke due to the structural form of its long guide rail. The general rehabilitation training robots only need to assist people in basic actions such as walking, and often do not needlarge acceleration. However, the transmission mode of this robot determines that it can realize large acceleration movement, especially in the direction of the guide rail (the movement in this direction is equivalent to the direct drive of the motors). Through software simulation, the acceleration can reach 5 m/s$^2$.

The kinematics and dynamics of the robot are analyzed to ensure the feasibility and reliability of the robot in practical applications. The structural and mechanical characteristics of the robot are analyzed, which lays a foundation for the subsequent development of the robot. The key algorithm of somatosensory simulation, the washout algorithm, is studied and optimized to make it more suitable for rehabilitation training. The performance improvements brought by optimization are verified by simulations.

The robot not only guarantees feasibility and reliability, but also provides excellent performance. It has broad market prospects and great application value.

The major contributions of this paper are as follows:

- Based on a detailed investigation in related fields, this paper designs and constructs a robot system to provide somatosensory simulation for rehabilitation training. This paper presents a new type of parallel platform with a large load, high acceleration and six degrees of freedom. The moving platform has a large range of motion, especially along the guide rail. It adopts a structure of double parallel use, which has the advantages of low platform height and high proportion of payload. The driving mechanism adopts the form of a parallel guide rail, which easily realizes high-speed and high-acceleration movement. A new structural form is proposed for the design of a vestibular simulation platform.

- The robot can be redeveloped because the height of the moving platform is low. Another set of mechanical devices can be connected in series on the moving platform to realize deep sensory somatosensory simulation. Thus, the function of the robot is further expanded.
- A comprehensive characteristic analysis of this new robot is carried out. We study the structural characteristics of the robot, optimize the overall structural parameters with the objective of reducing the driving force, improving the performance of the whole machine, and guiding the completion of the mechanical structure design which provides a theoretical and experimental basis for the application and verification of the subsequent washout algorithm.
- An improved washout control algorithm of a 6-DOF parallel robot based on the MOGA is proposed. The feasibility of the washout algorithm based on MOGA optimization is verified by simulations.

## 7. Patents

Liu Yubin, Wu Junyu, Zhao Jie, Man Zhuoqi. Somatosensory simulation parallel platform integrating human deep sensation and vestibular sensation. 202110155514.4 (under acceptance).

**Author Contributions:** Conceptualization, J.W. and Y.L.; software, J.W.; investigation, J.W. and X.Z.; data curation, J.W.; writing—original draft preparation, J.W.; writing—review and editing, J.W. and Y.L.; visualization, J.W.; supervision, Y.L and J.Z.; project administration, Y.L., J.Z. and Y.G. All authors have read and agreed to the published version of the manuscript.

**Funding:** The work described in this paper is supported by national key R & D program (Ministry of science and technology of China: 2019YFB1311400).

**Institutional Review Board Statement:** Not applicable.

**Informed Consent Statement:** Not applicable.

**Data Availability Statement:** All data are open source and can be found on the Internet.

**Conflicts of Interest:** The authors declare that the research was conducted in the absence of any commercial or financial relationships that could be construed as a potential conflict of interest.

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
