# Peer review of "Research on Theory and a Performance Analysis of an Innovative Rehabilitation Robot"

_sensors, doi:10.3390/s22103929_

Round 1

Reviewer 1 Report

The authors of this paper present a novel application of a 6-DOF robot in the field of rehabilitation training. Intended for lower limb rehabilitation, the proposed robot adopts a new structure that facilitates patients to carry out a variety of rehabilitation exercises. A new 6-DOF parallel platform with a large load and high acceleration is proposed. Its main target is to help patients that have balance disorders with somatosensory simulation and rehabilitation training. In order to realize a variety of motions in a limited range, the authors implement a washout control algorithm. To overcome the limitations of the classic washout algorithm, the MOGA algorithm is used to optimize different parameters.

In general, the paper is well-organized and having clear explanations. Some preliminary results verify the validity of the proposed manipulator.

Please review the English writing and correct some typos, such as:

  • Abstract: line 11, “can” is repeated twice
  • The last two keywords are the same ones.
  • Section 2, line 42, should say “and” not an
  • Line 153: a numerical solution… “based on”…
  • Line 279: is “closed” to

When starting the explanation of section 2.2, the authors should refer to Fig. 3, where the robot structure is shown. They don´t mention that figure until the end of the whole paragraph. Similarly, refer to Fig. 4 in section 3.1.

I recommend the authors to enlarge the state of the art and include more references, as in the field of rehabilitation many different and interesting works carried out by many authors could be additionally cited.

Reviewer 2 Report

This paper focuses on an interesting topic of designing a rehabilitation-assisted robot. As illustrated by the theoretical analysis and experimental verifications, the designed robot can not only guarantee feasibility and reliability, but also provides excellent performance on rehabilitation exercises. However, the following comments should be considered for improving this paper:

(1) Most of the figures are of low resolution. Please update them with suitable size and quality.

(2) Lacking of sufficient comparisons with the mentioned methods and mechatronics in related works.

(3) The references are incomplete, and their formats are not uniform.

Round 2

Reviewer 2 Report

The authors have addressed most of the comments.  In conclusion, the paper is worthy of research and can be accepted.

Author Response

Thank you for your comments.